# Enzymatic Hydrolysis of Orange-Footed Sea Cucumber (*Cucumaria frondosa*)—Effect of Different Enzymes on Protein Yield and Bioactivity

**DOI:** 10.3390/foods12193685

**Published:** 2023-10-07

**Authors:** Dat Trong Vu, Eva Falch, Edel O. Elvevoll, Ida-Johanne Jensen

**Affiliations:** 1Department of Biotechnology and Food Science, The Norwegian University of Science and Technology, NTNU Trondheim, N-7012 Trondheim, Norway; dat.t.vu@ntnu.no (D.T.V.); eva.falch@ntnu.no (E.F.); 2Norwegian College of Fishery Science, UiT-The Arctic University of Norway, N-9037 Tromsø, Norway; edel.elvevoll@uit.no

**Keywords:** seafood, marine organisms, in vitro biological activity, sea cucumber, enzymatic hydrolysis, functional food, peptide, amino acid

## Abstract

While sea cucumber is a food delicacy in Asia, these food resources are less exploited in Europe. The aim of this study was to determine the chemical composition and potential food applications of the less exploited orange-footed sea cucumber (*Cucumaria frondosa*). In particular, the antioxidative capacity and free amino acids associated with the umami flavor released by enzymatic hydrolyses by either Bromelain + Papain (0.36%, 1:1) or Alcalase (0.36%) were studied. Fresh *C. frondosa* contained approximately 86% water, and low levels of ash (<1%) and lipids (<0.5%). The protein content was 5%, with a high proportion of essential amino acids (43%) and thus comparable to the FAO reference protein. The high concentration of free amino acids associated with umami, sour, sweet, and bitter may contribute to flavor enhancement. Hydrolysis by Bromelain + Papain resulted in the highest protein yield, and the greatest concentration of free amino acids associated with umami and sour taste. All samples showed promising antioxidant capacity measured by FRAP, ABTS, DPPH and ORAC compared to previous reports. The inorganic arsenic concentration of fresh *C. frondosa* ranged from 2 to 8 mg/kg wet weight and was not affected by processing. This is comparable to other seafood and may exceed regulatory limits of consumption.

## 1. Introduction

Marine organisms are in general perceived as a healthier food choice compared to terrestrial organisms, contributing with omega-3 fatty acids, micronutrients, proteins as well as bioactive peptides [1,2]. In some populations, aquatic food provides as much as 50 percent of dietary animal protein, and globally, aquatic food provides about 17 percent of animal protein [3]. The harvest of lower tropic marine sources has been suggested as a strategy to contribute to food security [4], and a further shift in human diets from terrestrial protein sources, with higher footprints, to plant-based and marine lower-food-web sources is warranted to reduce dietary footprint [5]. A challenge to this shift has been suggested to be the lack of umami taste in vegetable foods [6]. A flexitarian diet may be a solution, as it is a moderate transition from meat-based diets to a greener diet and thereby reduces the consumption of animal food [7,8]. To safeguard the natural resources and prevent the depletion of commonly harvested fish stocks [3], underutilized marine resources such as sea cucumber, seaweeds and algae may be good alternatives [7,9,10,11].

Sea cucumbers are valued in Asia as a nutritious and luxury food component, and have been a part of the traditional Asian diet for generations [11,12,13]. Although not traditionally consumed in Nordic countries, it is harvested commercially and exported fresh or dried, or as nutraceutical and pharmaceutical extracts [11,12].

*Cucumaria frondosa* (*C. frondosa*) has a broad distribution in the frigid waters of the North Atlantic Ocean and near the shore of the Russian Federation in the Barents Sea. *C. frondosa* is the species of sea cucumber that is most prevalent in northern countries [11]. In North America and in Canada, *C. frondosa* has been harvested for decades [14]. The potential for commercial harvest in Norway was surveyed in 2018 [15,16]. However, no commercial harvest is ongoing yet [17]. In addition to nutrients, non-essential elements such as heavy metals may be present in sea cucumbers. Inorganic arsenic is present on the ocean bed and has negative effects on human health [18]. The content of inorganic arsenic in *C. frondosa* has been reported to vary with some individuals containing high levels [16,19] or levels exceeding regulatory limits of consumption [20]. Thus, the inorganic arsenic concentration should be monitored [16,18]. One way to exploit *C. frondosa* is to extract and utilize bioactive components, through the help of enzymatic hydrolysis [11,21]. Different food-grade enzymes [22] such as Bromelain, Papain or Alcalase, could be used to increase the bioavailability of small peptides that might contribute to an increase in antioxidant capacity [23]. Bromelain, Papain and Alcalase are commonly used in the food industry to assist the tenderization of meat, fish sauce production, baking, dairy product processing, and cheese processing [24,25,26]. Bromelain + Papain (1:1, 0.1%) has previously been used in combination to produce saithe (*Pollachius virens*) protein hydrolysates (pH 6, 50 °C, 60 min) [27]. In addition, Alcalase (4000 U/g substrate) has previously been used to produce hydrolysates from orange-footed sea cucumber (*C. frondosa*) viscera (pH 10, 50 °C, 60 min) [28].

Previous in vitro studies of *C. frondosa* have shown several bioactive properties [11], such as antioxidant capacities and antihypertensive properties, implying that this sea cucumber might be used as a potential functional food ingredient [21,29,30,31].

This study was conducted with the aim of analyzing the chemical composition of the underutilized orange-footed sea cucumber (*C. frondosa*) and exploring its potential food applications. In particular, the antioxidative capacity and free amino acids associated with umami released by enzymatic hydrolyses by either Bromelain + Papain (0.36%, 1:1) or Alcalase (0.36%) were studied.

## 2. Materials and Methods

### 2.1. Raw Material

*C. frondosa* (*n* = 5), weighting 480 ± 91 g, were caught by divers in February 2021 off the coast of Tromsø, (69° 35.795 N 18° 56.047 E) at a 15 m depth. Whole sea cucumbers were kept in seawater and frozen within a few hours, before being sent to NTNU and stored at −80 °C. *C. frondosa* was freeze-dried at −40 °C at 0.040 mBar using Labconco FreeZone 12 (Labconco Corporation, Kansas City, MO, USA, −50 °C, <13.3 Pa), homogenized to powder by a kitchen blender (Electrolux, ESB2600, 600 W 50 hz, Stockholm, Sweden) and stored at −80 °C for further analysis.

### 2.2. Study Design

The study design of *C. frondosa* consisted of processes such as freeze-drying, homogenization of freeze-dried sample and enzymatic hydrolysis (Figure 1). The comparison of freeze-dried samples was carried out in terms of proximate composition, minor constituents and antioxidative properties with hydrolysate sample.

### 2.3. Proximate Composition

Water and ash content were analyzed and calculated gravimetrically as described in the official method [32]. The protein content of freeze-dried sample was determined as described by the AOAC [33], while the method used to determine the protein content of the hydrolysates involved calculating the total sum of all amino acids, taking into account that the molecular weight of water was removed. This approach is in line with the guidelines set forth by the Food Agricultural Organization of the United Nations (FAO) [34]. Lipid was extracted and the total amount calculated gravimetrically as described by the Bligh and Dyer [35] method. The analysis was performed in triplicate.

### 2.4. Enzymatic Hydrolysis

The enzymatic hydrolysis was performed as described in Hjellnes et al. [27] with modification. Freeze-dried *C. frondosa* powder (5.0 ± 0.0 g) was mixed with preheated water (50 °C), a ratio between sample/water of 1:6.5 for 60 min at 50 °C with magnetic stirring (300 rpm). Bromelain (*Ananas comosus*, 3 U/mg, EC 3.4.22.32, Merck Life Science AS, Oslo, Norway), and Papain (*Carica papaya*, 1.5–10 U/mg, EC 3.4.22.2, Merck Life Science AS, Oslo, Norway) were added in a 1:1 ratio, and Alcalase (from Bacillus licheniformis, ≤2.4 U/g, EC 232.752.2, Sigma-Aldrich, Steinheim, Germany) accounted for 0.36% of the total mixture. The enzymatic hydrolysis mixture was added in glass bottles with a blue cap in a water bath (Heto-Holten, Allerød, Denmark). The enzymatic reaction was stopped by applying heat (90 °C, 20 min). The mixture was then distributed into 15 centrifuge tubes (50 mL) and subjected to centrifugation (3500 g, 30 min, 20 °C). After this, the samples were frozen (−20 °C). The frozen samples were then divided into three fractions: oil, hydrolysate, and sludge, using a scalpel. The protein hydrolysates from *C. frondosa* were subsequently freeze-dried using a Labconco FreeZone 12 (Labcono Corporation, Kansas City, MO, USA, at −50 °C, <13.3 Pa). These samples were then stored at −80 °C for future analysis. The enzymatic hydrolysis was performed in duplicate.

### 2.5. Total Amino Acids

The total amino acids content was determined as described by Blackburn [36]. Freeze-dried *C. frondosa* (87.5 ± 7.3 mg) and hydrolysate material (49.7 ± 0.3 mg) had 6 M HCl (SigmaAldrich, Darmstadt, Germany) added. The samples underwent hydrolysis for a period of 22 h at a temperature of 105 °C. The samples were neutralized to a pH range of 6.5–7.5 using 6 M NaOH (SigmaAldrich, Darmstadt, Germany), filtered using a Whatman glass microfiber filter GF/C (Cytiva, Washington, DC, USA), diluted 1:500 and passed through a 0.22 µm syringe filter (VWR, Radnor, PA, USA). The filtered samples were then analyzed using an HPLC, Dionex Ultimate 3000 (Dionex, CA, USA) with a Nova-pak c184 μM 3.9 × 150 mm column (Thermo Fischer Scientific, Waltham, MA, USA). The analysis was conducted using a fluorescence detector (Rf 200). Diluent phase A is methanol (VWR, Rosny-sous-Bois, France) and diluent phase B is sodium acetate (Alfa Aaser, Kandel, Germany) with flow rate 0.9 mL/L. Alpha-aminobutyric acid served as the internal standard in this analysis. The analysis was repeated two times for each sample.

### 2.6. Free Amino Acids

The content of free amino acids was assessed as described in Osnes and Mohr [37]. Water-soluble proteins were extracted (0.1 ± 0.0 g dry weight in 5 mL dH_2_O) by mixing and centrifuging freeze-dried or hydrolysate material of *C. frondosa*. Water-soluble protein extracts (1 mL) were then transferred to microcentrifugation tubes and added sulphosalisylic acid (0.25 mL, 10%, VWR, Oslo, Norway). The samples were vortexed, left in a cold room (4 °C) for 30 min before centrifuging for 10 min at 18,000× *g*. The supernatant was diluted 1:25, filtered and analyzed using HPLC, Dionex Ultimate 3000 (Dionex, CA, USA). The same system parameters were applied as in the Total amino acid method. The analysis was repeated two times for each sample.

### 2.7. Inorganic Arsenic Analysis

The concentration of inorganic arsenic was examined using HPLC-ICPMS by an accredited external laboratory (Institute of marine research, Bergen) [38,39,40]. Pooled sampling was done on hydrolysates without enzyme, Bromelain + Papain and Alcalase. The analysis was performed five times for freeze-dried *C. frondosa* and one time for pooled sampling for hydrolysates without enzyme, hydrolysate using Bromelain + Papain and hydrolysate using Alcalase.

### 2.8. Molecular Weight Distribution

The analysis of the molecular weight distribution for both the freeze-dried and hydrolysate materials was conducted using High-Performance Liquid Chromatography (HPLC), following a method that has been previously described [27]. The analysis was performed one time for each sample.

### 2.9. Antioxidant Assays

The antioxidant activity of both freeze-dried and hydrolysate material were analyzed by 2′-azino-bis(3-ethylbenzothiazoline-6-sulfonic acid (ABTS), 2,2-diphenyl-1-picrylhydrazyl (DPPH), ferric reducing antioxidant power (FRAP) and oxygen radical absorbance capacity (ORAC). ABTS was performed as described in Re et al. [41] and Nenadis et al. [42,43]. The ABTS assay measures the ability of a compound to neutralize reactive oxygen species (ABTS^+•^ radical, green-blue color) by electron transfer, to a stable ABTS molecule (pale color). The paler the color gets, the higher the antioxidant capacity of the sample [41,42,43]. DPPH was carried out as previously described by Thiansilakul et al. [44] and Nenadis et al. [43]. The DPPH method measures the ability of a compound to neutralize reactive oxygen species (DPPH^+•^ radical, dark purple color) by hydrogen transfer, to a stable DPPH molecule (pale yellow color). The paler the color gets, the higher antioxidant capacity of the sample [43,44,45]. FRAP was conducted as previously described by Benzie and Strain [46] with modifications described by Jensen et al. [47]. The FRAP method measures the ability of a compound to reduce ferric (Fe^3+^) to ferrous (Fe^2+^) iron by electron transfer (pH 3.6) and the sample solution would result in a deep blue color due to the formation of the ferrous–tripyridyltriazine complex [46]. ORAC was performed as described by Dávalos et al. [48] with modifications described by [47]. The ORAC assay is based on the oxidation of a fluorescent probe due to the free radical initiator (2,2′-azobis-(isobuttersa ureamidin)-dihydroclorid (AAPH)) which in turn leads to a decay of the fluorescence probe. Thus, the ORAC assay measures the ability of a compound to delay the oxidation of the fluorescent probe by reacting with peroxyl radicals by hydrogen transfer [48]. Trolox (Acros Organics, Geel, Belgium) was used as the reference compound for FRAP and ORAC. Propyl gallate (Sigma-Aldrich, Steinheim, Germany) was used as the reference compound for ABTS and DPPH. The results were presented in terms of μmol/g propyl gallate equivalents or μmol/g Trolox equivalents, respectively. The analysis was repeated two times for each sample.

### 2.10. Statistical Analysis

All results are presented as average ± standard deviation of *n* = 5 parallels, unless stated otherwise. Significant differences were assessed using Microsoft Excel and IBM SPSS Statistics, version 27 (IBM, New York, NY, USA) between hydrolysates with different enzyme. It was assumed that all data were normally distributed, and ANOVA was conducted with Tukey’s post hoc test.

## 3. Results and Discussion

### 3.1. Proximate Composition of C. frondosa

The biochemical composition of *C. frondosa* (Table 1) was comparable to previous studies, except for ash content which was lower when compared to previous results [49,50]. The ash content is reported to vary with mineral deposits in sea cucumber [51].

### 3.2. Total and Free Amino Acid Composition of C. frondosa

The amino acids glutamic acid, glycine/arginine and aspartic acid together constituted 43% (130 mg/g dry weight) of the total amino acid of *C. frondosa* (305 mg/g dry weight, Table 2). This is comparable to previously findings by Zhong et al. [49].

Glutamic acid and glycine/arginine, together with alanine were also the dominating free amino acids, constituting about 78% (5.8 mg/g dry weight) of the free amino acid raw material (7.4 mg/g dry weight, Table 2). This is comparable to previous studies by Song et al. [52]. Furthermore, in the study by Sroyraya et al. [53] on sea cucumber *Holothuria scabra*, glutamic acid made out the highest level of the free amino acids analyzed. Glycine/Arginine and alanine were also previously reported to be dominating. It should, however, be emphasized that habitat location and seasonal variations could influence the concentration of these compounds [54].

### 3.3. Protein Content and Protein Yield of C. frondosa Hydrolysates

The protein content in the hydrolysates was determined by calculating the sum of amino acids, taking into account that the molecular weight of the water was removed [34]. Hydrolysate from the use of Bromelain + Papain resulted in the highest protein yield followed by hydrolysate using Alcalase and hydrolysate without additional enzyme (Table 3). The hydrolysate using Bromelain + Papain showed a significantly higher (*p* < 0.05) protein content compared to freeze-dried raw material, whereas the protein contents in hydrolysates using Alcalase and without enzyme treatment gave a lower protein content compared to freeze-dried sample.

Papain from *Carica papaya* has an optimum pH and temperature between 6.0 to 7.0 and 65 °C, respectively. The enzyme has a specific activity at 10 units/mg protein [55]. Papain mainly cleaves peptide bonds consisting of arginine, lysine and residues succeeding phenylalanine [55,56,57]. Bromelain from pineapple stem has an optimum pH and temperature at 7.0 and 50 °C, respectively. The enzyme has a specific activity around 3 units/mg protein [58]. Bromelain cleaves at the carbonyl end of glycine, tyrosine, alanine and lysine [59]. Alcalase from *Bacillus licheniformis* is active between pH 6.5 and 8.5 and has an optimum temperature at 60 °C [60]. The enzyme has a specific activity 2.4 × 10^−3^ units/mg protein. Alcalase cleaves peptide bonds on the carboxyl side of glutamine, lysine, tyrosine, leucine, methionine, and glutamic acid [61].

The enzymatic hydrolysis of *C. frondosa* using Bromelain + Papain resulted in a higher protein yield (36.8 ± 6.6%) compared to the study by Slizyte et al. [62] on fish protein hydrolysates from defatted salmon backbones, which found Bromelain + Papain to have 17% protein yield, however, with a higher enzyme substrate ratio.

In this study, all the enzymatic hydrolysis was performed at temperature 50 °C which may explain why enzymatic hydrolysis using Bromelain + Papain resulting in a higher protein yield, thus giving it a higher protein content, compared to enzymatic hydrolysis using Alcalase and enzymatic hydrolysis without enzyme. Hence, the Alcalase was outside its optimum temperature. In addition, this study does have certain limitations, including an incomplete total amino acid profile. The lack of cysteine and tryptophan in the assessment leads to an underestimation of the protein content.

### 3.4. Total and Free Amino Acid Composition of C. frondosa Hydrolysates

The major amino acids of all three hydrolysates were glutamic acid, glycine/arginine, and aspartic acid, constituting from 44 to 52% of the total amino acids (Table 4). This is comparable to previous findings [21,28].

The major free amino acids were the same, glutamic acid, glycine/arginine, and alanine, in all hydrolysates. Together, these amino acids constituted about 54% of the free amino acids in hydrolysate using Bromelain + Papain, 68% in hydrolysate using Alcalase and 74% in hydrolysate without enzyme (Table 4). The results in this study are similar to a previous study [63].

The enzymatic hydrolysis using Bromelain + Papain gave a higher total amino acid content compared to hydrolysate using Alcalase and hydrolysate without enzyme. This could be explained by the optimum condition of the different enzymes as previously described.

In the amino acid determination established by Blackburn [36], the amount of amino acid residues in the sample are analyzed after acid hydrolysis of peptide bonds, and the protein content is determined by adding up the individual amino acid residues, after subtracting the molecular mass of water [64]. This calculation aligns with the guidelines provided by the Food and Agricultural Organization of the United Nations (FAO) for protein determination in food [34]. The process efficiently hydrolyses most of the peptide bonds; however, the protein content may be underestimated due to the destruction or reduction of some amino acids prior to the HPLC analysis [64]. Thus, tryptophan and cysteine were not evaluated due to their instability during the process of acid hydrolysis. As a result, if the protein content is to be calculated based on the total amino acid content, these two amino acids should be analyzed separately and added on.

### 3.5. Protein Quality

The Food and Agriculture Organization of the United Nations (FAO) and World Health Organization (WHO) has established recommended intake values for essential amino acids, which are compared to a reference protein [65]. These values are based on the mean nitrogen requirement which should be sufficient to maintain body nitrogen homeostasis in healthy adults [65]. The concentration of essential amino acids in freeze-dried sample and hydrolysate without enzyme, hydrolysate using Bromelain + Papain or Alcalase (dry weight, Figure 2) were higher when compared to the reference protein (chemical score >1.0) [65]. Furthermore, freeze-dried sample, hydrolysate without enzyme, hydrolysate using Bromelain + Papain or Alcalase (dry weight) contained a higher concentration of essential amino acids compared to the findings of a previous study conducted by Kendler et al. [66] on flounder, lemon sole, megrim plaice and thornback ray. In addition, the levels of essential amino acids (dry weight) in this study were higher compared to the essential amino acids content of conventional Atlantic salmon [67], indicating that both freeze-dried and hydrolysates of *C. frondosa* could serve as excellent sources of essential amino acids due to its superb quality proteins.

The protein content of raw material based on wet weight is lower compared to other conventional seafood products (Table 5). However, freeze-dried sample, hydrolysate without enzyme, and hydrolysate using Bromelain + Papain or Alcalase (dry weight) have similar or higher protein content compared to the conventional seafood products, indicating that there is a high quality of proteins in *C. frondosa* after processing. In addition, both freeze-dried *C. frondosa* and the hydrolysates (dry weight) contained a higher concentration of essential amino acids compared to ‘Honeycrisp’ apple flesh [68] and banana (*Grand niene*) [69]. It is worth mentioning that tryptophan was not detected in the study by Wang et al. [68], Gurav and Jadhav [69] or in this present study.

### 3.6. Umami

The free amino acids of freeze-dried and hydrolysate without enzyme, hydrolysate with Bromelain + Papain and hydrolysate with Alcalase of *C. frondosa* (Figure 3) have been categorized based on their flavor profiles, as described in the studies by Sarower et al. [71], Kirimura et al. [72] and Fuke and Konosu [73], and as presented in Kendler et al. [66]. According to the study by Wang et al. [74], glutamic acid, glycine and alanine are identified among the most important umami taste contributors in seafood.

In the ‘sweet’ group, the concentration of glycine/arginine ranked highest of free amino acids followed by alanine and lysine. Serine from freeze-dried sample was significantly lower (*p* < 0.05) compared to the hydrolysates. In the ’bitter’ group, the free amino acid concentration of phenylalanine, tyrosine, valine, methionine, and histidine varied. All amino acids, apart from histidine in the ‘bitter’ group of hydrolysates using Bromelain + Papain were significantly different (*p* < 0.05) compared to freeze-dried sample, hydrolysate without enzyme and hydrolysate using Alcalase. In the ‘umami/sour’ group, glutamic acid represented the highest free amino acid concentration followed by aspartic acid. In addition, compared to other hydrolysates and the freeze-dried sample, the aspartic acid content of the hydrolysate produced using Alcalase was significantly higher (*p* < 0.05). When sodium salts like monosodium glutamate (MSG) are present, these amino acids may contribute to the umami flavor [66,74].

The concentration of free amino acids (dry weight), which could contribute to sweet, umami and sour flavor in this study, exceeded the free amino acid concentration in Atlantic salmon (dry weight) up to 3 times [75], indicating there is a high concentration of such flavor components in freeze-dried and hydrolysate samples. In addition, the concentration of free amino acids contributing to bitter flavor was 20 times less in this study compared to Atlantic salmon (dry weight) [75]. However, the composition and properties of small peptides and umami free amino acids play an important role eliciting the characteristic taste of food [66,71,76].

According to the study by Hoehl et al. [77] on recognition threshold for basic taste in deionized water, 70 females from Germany recognized the flavor of sweet, salty bitter and umami at 0.2, 0.05, 0.08 and 0.09 mg/L, respectively. In this study, the result exceeded the minimum requirements to taste sweet, salty, bitter and umami. Therefore, the freeze-dried and hydrolysates of *C. frondosa* could potentially be used as a “condiment” and contribute with the umami taste that is lacking from vegetable foods. This could ease the transition from a meat-based diet to a greener diet and potentially reduce the consumption of animal food. However, in this study, it is important to note that umami and sour flavor could not be separated. Additionally, it is vital to remember that the threshold of perception of taste varies with age, sex and body mass in different countries with different food and cultures [78].

### 3.7. Total Inorganic Arsenic Concentration

An important prerequisite before novel resources can be used, is an assessment of food safety [20]. Although the organic compound of arsenic is generally present in seafood, the inorganic compounds of arsenic may be abundant in some organisms. Therefore, it is necessary to specify each product as some sea cucumbers are an exception to this rule [20].

The freeze-dried and the hydrolysates samples were analyzed for inorganic arsenic concentration, varying from 4.1 to 7.8 mg/kg wet weight. Protein hydrolysates had higher content of inorganic arsenic compared to freeze-dried sample (wet weight) (Figure 4). The results in this study are comparable to previous results [79] (pelagic, 0.01; molluscs, 0.08 and demersal, 0.008 mg/kg wet weight), but contained a higher level of inorganic arsenic. In addition, the inorganic arsenic concentration did not change in the hydrolysate during the freeze-drying process. The same result was observed in the study by Dahl et al. [80] on stability of arsenic compounds in seafood during processing and storage by freezing. However, these results must be verified as only one pooled sample was analyzed due to limited availability of raw materials.

The World Health Organization [81] established a limit for human dietary exposure of seafood. The Provisional Tolerable Weekly Intake (PTWI) for inorganic arsenic was set to 3 µg/kg body weight [81]. An adult with a body weight of 70 kg should hence not exceed an intake of 0.21 mg inorganic arsenic. Based on the inorganic arsenic content in the samples analyzed in this study, it refers to between 27 to 51 g sea cucumber (dry weight). The level of inorganic arsenic may limit the applicability of *C. frondosa* as a potential food ingredient. However, it is crucial to note the low number of samples analyzed in this study.

A recent review on sea cucumbers and food safety [20] documented high levels of inorganic arsenic concentration, depending on its location [81]. Sea cucumbers live on different habitats on the ocean floor which leads to a different accumulation of heavy metals [12]. A previous study has shown that the content of inorganic arsenic in seaweed was reduced by boiling and soaking in salt solution [82]. This is due to arsenic being quite water-soluble, and soaking with an increase in salt concentration, the water content will decrease; thus, the total arsenic concentration will decrease [82]. In another study by Lin et al. [83], heavy metals such as arsenic and lead were removed from sea cucumber *Acaudina leucoprocta* by first hydrolyzing the body wall mucus protein with Papain and pepsin (0.15% (1:1, *w*/*w*) at 37 °C for 45 min), and then removing the heavy metals through immersing in citric acid (0.1 M for 54 h). *Acaudina leucoprocta* produced a product that had a darker, more opaque appearance and a looser body wall structure. About 98.22 ± 0.91% of the arsenic and 94.11 ± 1.08% of the lead were eliminated after 54 h of immersing. The heavy metal concentrations were < 0.5 mg/kg in all samples. This could maybe be the solution to reduce the total arsenic concentration in *C. frondosa* and increase its applicability as a food ingredient. This should be further investigated before processing *C. frondosa*.

### 3.8. Molecular Weight Distribution

The distribution of molecular weights in freeze-dried and hydrolysate using Bromelain + Papain, hydrolysate using Alcalase and hydrolysate without enzyme of *C. frondosa* ranged between <0.2 and >20 kDa (Figure 5). Hydrolysate using Bromelain + Papain resulted in the highest concentration of peptides between <0.2 and 5–10 kDa, while hydrolysate using Alcalase resulted in the highest concentration of peptides between 10 to >20 kDa. Hydrolysate without enzyme resulted in a higher amount of <0.2 kDa peptides, although significantly less (*p* < 0.05) compared to hydrolysates with enzymes. The concentration of peptides with molecular size between 1–2 kDa of hydrolysate using Bromelain + Papain was significantly higher (*p* < 0.05) compared to hydrolysate using Alcalase, hydrolysate without enzyme and freeze-dried *C. frondosa*. Hydrolysate using Alcalase had a significantly higher peptide concentration (*p* < 0.05) compared to freeze-dried *C. frondosa*, hydrolysate without enzyme and hydrolysate using Bromelain + Papain with a molecular size between >10 and >20 kDa. In addition, as expected, the peptide concentration of freeze-dried *C. frondosa* was low compared to the hydrolysates.

There are several explanations why the peptide concentration is lower in samples with higher protein content. The first explanation could be due to the selection and the limited detection range of the separation column. Not all peptides could be detected or there could be other responses in the detector from the original protein. In this study, a column with a detection range between 0 and 20 kDa was selected. The second explanation could be due to incomplete enzymatic hydrolysis, or that the peptide chains are too long to be detected.

The peptide’s ability for antioxidant potential obtained from enzymatic hydrolysis of sea cucumber depends on the molecular length and weight of the peptide (≤20 kDa), the amino acid composition and the sequences [84]. In addition, the presence of amino acids such as arginine, histidine, glutamic acid, tyrosine, phenylalanine, and proline may improve the antioxidant activity of the bioactive peptides. Furthermore, in order to acquire bioactive peptides and antioxidant efficacy, the choice of proteases used in the production of protein hydrolysates is very important [84]. The formation of smaller peptides is dependent on the enzyme’s optimal conditions during enzymatic hydrolysis as previously described.

### 3.9. Antioxidative Capacity Measured by FRAP, ABTS, DPPH and ORAC Assays

The FRAP, ABTS, ORAC, and DPPH analyzes were utilized to assess the antioxidative ability [85]. The FRAP method measures the ability of a compound to reduce ferric (Fe^3+^) to ferrous (Fe^2+^) iron by electron transfer (pH 3.6) and the sample solution would result in a deep blue color due to the formation of the ferrous–tripyridyltriazine complex [46]. The ferric reducing antioxidant power of the hydrolysates ranged between 14–26 µmol Trolox equivalents/g dry weight (Figure 6). Hydrolysate without enzyme and hydrolysate using Alcalase exhibited similar FRAP values, which was higher compared to the freeze-dried sample. In a previous study on silkworm pupae (*Bombyx mori*) [86], hydrolysate using Alcalase showed a higher FRAP value compared to hydrolysate without enzyme. The results in this study were comparable to previous studies [27,87,88]. Protein hydrolysate of red algae (*Palmaria palmata*) exhibited 1–22 µmol Trolox equivalents/g [87], and protein hydrolysate of saithe (*Pollachius virens*) exhibited 5–16 µmol Trolox equivalents/g [27]. In addition, the ferric reducing antioxidant power in this present study is higher compared to protein hydrolysate of fried herring (*Clupea harengus*), which was found to be 3 µmol Trolox equivalents/g [88]. However, in a recent study [89] on protein isolates of orange-footed sea cucumber (*Cucumaria frondosa*), the Trolox equivalents exhibited 80–530 µmol /g. The reason for this difference is probably due to the isolation of peptides, resulting in a higher concentration [89,90], compared to the hydrolysates obtained from the present study.

The ABTS assay measures the ability of a compound to neutralize reactive oxygen species (ABTS^+•^ radical, green-blue color) by electron transfer, to a stable ABTS molecule (pale color). The paler the color gets, the higher antioxidant capacity of the sample [41,42,43]. The ABTS radical scavenging activity ranged between 2–13 µmol Propyl gallate equivalents/g dry weight between the samples (Figure 6). Hydrolysate using Bromelain + Papain showed the highest antioxidant capacity followed by hydrolysate using Alcalase, hydrolysate without enzyme and freeze-dried *C. frondosa*, (*p* < 0.05).

The result in this study is comparable to a previous study [91]. Different extracts of far-eastern sea cucumber (*Stichopus japonicus*) exhibited an ABTS radical scavenging activity between 7–16 Trolox equivalents/g dry weight [91]. However, protein hydrolysate of saithe (*Pollachius virens*) exhibited 42–63 µmol Propyl gallate equivalents/g [27], and protein hydrolysate using Alcalase on different body parts of *C. frondosa* exhibited between 21 to 79 μmol Trolox equivalents/g protein [21]. The reasons for those differences may be due to the use of different species [27], standards [91] and the concentration of peptides [92], as compared to the hydrolysate obtained in the present study.

The DPPH method measures the ability of a compound to neutralize reactive oxygen species (DPPH^+•^ radical, dark purple color) by hydrogen transfer, to a stable DPPH molecule (pale yellow color). The paler the color gets, the higher antioxidant capacity of the sample [43,44,45]. The DPPH radical scavenging activity ranged between 2 and 5 µmol Propyl gallate equivalents/g dry weight. Hydrolysate using Alcalase exhibited the highest DPPH value, followed by hydrolysate using Bromelain + Papain, hydrolysate without enzyme and freeze-dried *C. frondosa* (Figure 6). Hydrolysate using Alcalase was found to exhibit a significant higher (*p* < 0.05) DPPH value compared to freeze-dried raw material, hydrolysate without enzyme and hydrolysate using Bromelain + Papain.

The exhibited DPPH scavenging activities of freeze-dried raw material and all hydrolysates were comparable to previous studies [21,49,93]. Coix seed fermented by *Monascus purpureus* exhibited 0.4 to 3.3 μmol Trolox equivalents/g [93], protein hydrolysate on different body parts of *C. frondosa* exhibited 7 to 16 μmol Trolox equivalents/g protein [21], and fresh and processed *C. frondosa* exhibited 5 to 7 μmol Trolox equivalents/g dry sample [49]. However, the wild bilberry (*Vaccinium myrtillus* L.) and lingonberry (*Vaccinium vitis-idaea* L.) leaves from north-west Romania exhibited 28 mM Trolox equivalents/g and 32 mM Trolox equivalents/g, respectively, which result in a higher DPPH activity [94] compared to the result on this present study. It is worth mentioning, the previous study used different species, standard and methodology to measure DPPH activity. This may account for the variations in DPPH activity compared to the findings in this present study.

The ORAC assay is based on the oxidation of a fluorescent probe due to the free radical initiator (2,2′-azobis-(isobuttersa ureamidin)-dihydroclorid (AAPH)) which in turn leads to a decay of the fluorescence probe. Thus, the ORAC assay measures the ability of a compound to delay the oxidation of the fluorescent probe by reacting with peroxyl radicals via hydrogen transfer [48]. The antioxidant capacity ranged between 1900–12,000 µmol Trolox equivalents (TE)/g dry weight (Figure 7). Hydrolysate using Bromelain + Papain showed the highest antioxidant capacity followed by hydrolysate using Alcalase, hydrolysate without enzyme and freeze-dried *C. frondosa* (5700, 4000 and 1900 µmol TE/g dry weight, respectively). Significantly differences (*p* < 0.05) of ORAC value were found in all sample groups except between sample hydrolysate without enzyme and Alcalase.

The oxygen radical absorbance capacity in this present study is higher compared to previous studies. Protein hydrolysate of saithe (*Pollachius virens*) exhibited 331 to 558 µmol Trolox equivalents/g [27], protein hydrolysate of Atlantic salmon trimmings exhibited 20 µmol Trolox equivalents/g [95], *C. frondosa* extract exhibited 140 to 800 µmol Trolox equivalents/g, and protein hydrolysate of red algae (*Palmaria palmata*) exhibited 45 to 168 µmol Trolox equivalents/g [87]. The differences could be due to different methodology to measure the oxygen radical absorbance capacity. Another explanation could be the different parts of *C. frondosa* used to measure oxygen radical absorbance capacity.

Hydrolysate using Bromelain + Papain appears to exhibit the highest antioxidant activity (followed by hydrolysate using Alcalase and without enzyme), while freeze-dried *C. frondosa* in general exhibited lower antioxidative capacity. It has previously been documented that some amino acids exhibit antioxidative activity. Amino acids such as proline, methionine, leucine, glycine histidine, tyrosine, lysine, cysteine, tryptophan and arginine showed a greater total antioxidant capacity compared to other amino acids [96,97]. Hydrolysate using Bromelain + Papain contained the highest concentration of leucine, tyrosine, histidine, methionine, and phenylalanine, followed by hydrolysate using Alcalase, hydrolysate without enzyme and freeze-dried sample, based on the content of free amino acids. This could explain the high antioxidant capacity of hydrolysate using Bromelain + Papain compared to other samples. However, proline, cysteine, tryptophan could not be detected, and arginine and glycine could not be separated in this study.

Both tyrosine and tryptophan can scavenge free radicals by electron donation [96]. In addition, the indole sidechain of tryptophan could contribute to the chelation of prooxidant metal ions [98]. The sidechain of cysteine and methionine could both act as an antioxidant where reactive oxidative species oxidizes the sidechain by electron donation [99,100]. Arginine and lysine both have free radical scavenging properties [100,101]. Moreover, the imidazole ring of histidine could chelate prooxidant metal ions and prevent the generation of radicals [102].

Hydrolysate using Bromelain + Papain exhibited the highest ABTS and ORAC activity compared to other samples, suggesting that the high content of specific free amino acids contributes to antioxidant activity as previously described. However, hydrolysate using Bromelain + Papain resulted in a lower FRAP and DPPH activity compared to other samples. A previous study reported hydrolysate using Papain resulted in low values of FRAP activity [103], while hydrolysate using Bromelain resulted in a higher FRAP activity, suggesting that hydrolysate using Papain could be due to the lower ability to reduce ferric ion to its ferrous form. However, other studies have found the opposite; hydrolysate using Papain resulted in a higher FRAP activity compared to hydrolysate using Bromelain [104,105]. It could be other mechanism involve of the antioxidant activity in the hydrolysate.

The FRAP and the ORAC activity determined in this study result in a higher antioxidant capacity compared to previous studies, indicating a potential use to contribute to food preservation. However, more studies are needed as the DPPH and ABTS radical scavenging activity value resulted in a lower radical scavenging activity compared to previous studies.

The analyses produced substantial variances in the observed antioxidant activity assays. According to the study by Hjellnes et al. [27] on Saithe (*Pollachius virens*), it could be challenging to evaluate results from different antioxidant assays both between and within assessments. The comparison of antioxidant capacity between these methods (FRAP, ABTS, ORAC, and DPPH) is difficult due to different use of standard (e.g., Propyl gallate and Trolox equivalents). Furthermore, methods are conducted between different conditions such as temperature, pH and solvents, and are measuring different mechanisms [106]. In addition, the degree of hydrolysis, the selection of enzymes used for hydrolysis, the size and solubility of the peptides, and the number of free amino acids that have antioxidant ability determine the antioxidant activity [29,107]. In this study, the molecular weight of peptides in the majority for all samples, was found to be in the region of <0.2 kDa. According to the studies by Je et al. [108] and Šližytė et al. [109] on hydrolyzed fish proteins, the antioxidant activity may be associated with a high concentration of small peptides. Most of the small peptides from freeze-dried *C. frondosa*, hydrolysate using Alcalase and without enzyme lies in the size <0.2 kDa, except for hydrolysate using Bromelain + Papain where it had the highest concentration of smaller peptides between 1–2 kDa. The results from ABTS activity regarding the peptide size in this study agrees with what was found in a previous study [110]. The same result was found for ORAC activity of hydrolysate using Bromelain + Papain, where the content of smaller peptides gave a greater ability to donate electrons and interact with the free radical compared to the larger peptides.

The result in that study shows the orientation of the molecular size which could contribute to the antioxidant capacity lies between 0 to 5 kDa. High levels of smaller peptides were found with a peptide size <1 kDa. This is in accordance with previous studies showing that the concentration of the smaller peptides in the hydrolysates produced from enzymatic hydrolysis exhibit antioxidant properties [111,112,113]. The antioxidant activity based on FRAP, ABTS, DPPH and ORAC may not be primarily acting on the high concentration of smaller peptides [27]. It may also be reliant on several features, such as the interaction between the bigger peptides and the smaller peptides of the sample. Furthermore, it can also rely on the sea cucumber species, harvesting region, dietary preference, body part, and processing techniques used [84]. The antioxidant capacity could be linked to the smaller peptides. In addition, no clear connection was found between high amino acid content and high levels of smaller peptide sizes gave higher antioxidant capacity. A suggestion would be to do more studies at this topic.

## 4. Conclusions

This study investigated the chemical composition and potential food applications of orange-footed sea cucumber (*Cucumaria frondosa*). In particular, the antioxidative capacity and free amino acids associated with umami released by enzymatic hydrolyses by either Bromelain + Papain (0.36%, 1:1) or Alcalase (0.36%) were studied.

Despite low protein content (5%) in fresh *C. frondosa*, it is of high quality. The essential amino acids were high compared to the reference protein (chemical score > 1.0), and the high concentration of free amino acids associated with umami, sour, sweet and salty may contribute as a flavor enhancer. Hydrolysate using Bromelain + Papain provided the highest protein yield and the highest antioxidant activity, probably explained by the fact that the enzymatic hydrolysis was performed at optimum temperature for these enzymes.

The inorganic arsenic concentration of fresh *C. frondosa* was not affected by processing. Due to a limited number of samples, these results must be verified.

The hydrolysate of *C. frondosa*, produced by the addition Bromelain and Papain, exhibited a promising antioxidant capacity as indicated by the ferric reducing antioxidant power (FRAP) and oxygen radical absorbance capacity (ORAC). However, 2′-azino-bis(3-ethylbenzothiazoline-6-sulfonic acid (ABTS) and 2,2-diphenyl-1-picrylhydrazyl (DPPH) radical scavenging activities were found to be lower compared to what has been observed in previous studies. This study demonstrates that *C. frondosa* may provide interesting functional properties such as taste enhancement and antioxidant capacity, and be a potential functional ingredient; however, more studies are recommended to verify the antioxidant capacity in *C. frondosa*.

## Figures and Tables

**Figure 1 foods-12-03685-f001:**
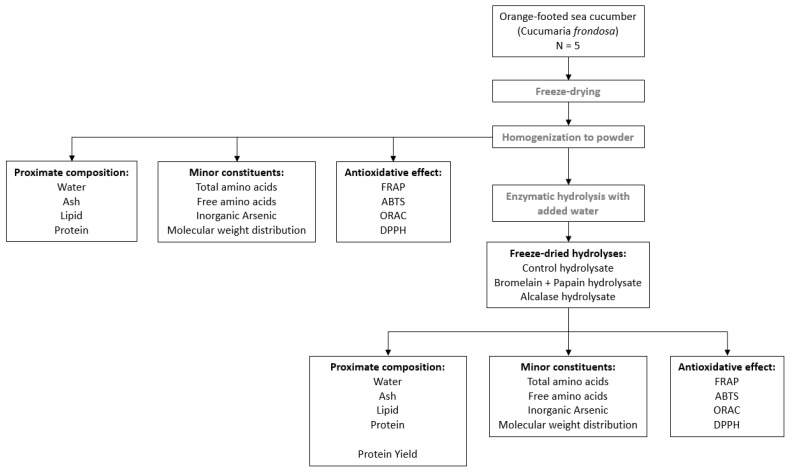
Schematic overview of analyses performed to characterize freeze-dried and hydrolysate samples of *Cucumaria frondosa*.

**Figure 2 foods-12-03685-f002:**
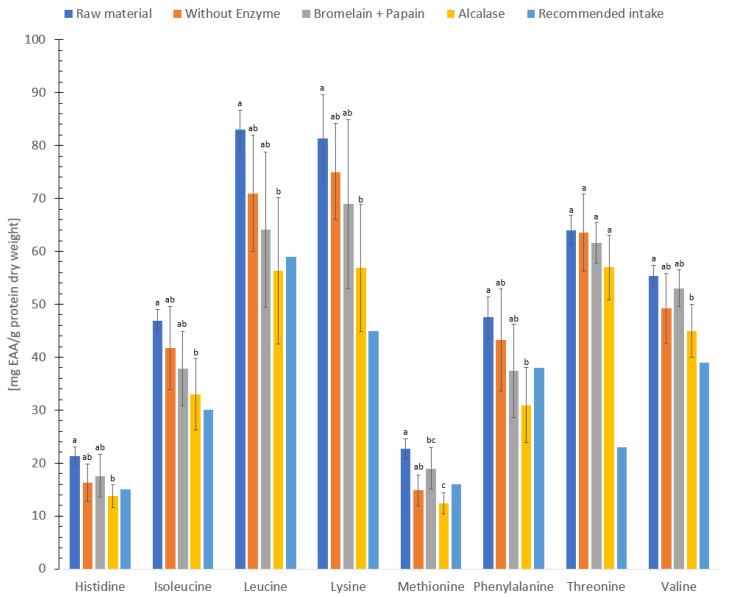
Concentration of essential amino acid (EAA, mg/g protein) in freeze-dried raw material and hydrolysates of *Cucumaria frondosa*, compared to a reference protein [65]. The results are presented as mean ± standard deviation (mg essential amino acid/g protein dry weight, *n* = 5). Values with different letters are significantly different (*p*  <  0.05).

**Figure 3 foods-12-03685-f003:**
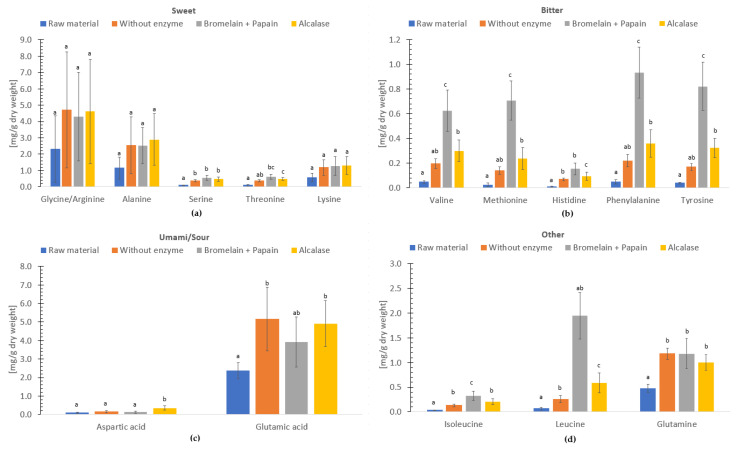
Free amino acids (mg/g dry weight) in freeze-dried raw material, and hydrolysates of *Cucumaria frondosa*, without enzyme, with Bromelain + Papain and with Alcalase. The distribution is grouped in taste perceptions sweet (**a**), bitter (**b**), umami/sour (**c**) and other (**d**). The results are presented as mean ± standard deviation (*n* = 5). Columns with distinct lower case letters within the same amino acids are significantly different (*p*  <  0.05).

**Figure 4 foods-12-03685-f004:**
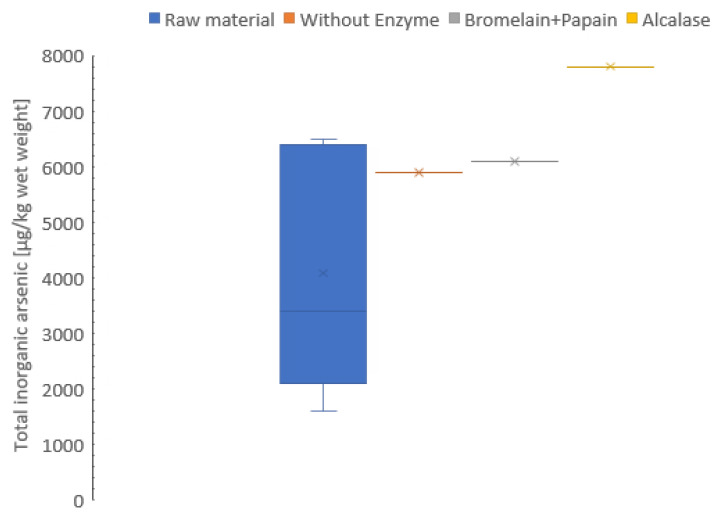
Inorganic arsenic (µg/kg wet weight) of freeze-dried raw material and hydrolysates of *Cucumaria frondosa*, protein hydrolysate without added enzyme, hydrolysate using Bromelain + Papain, and hydrolysate using Alcalase. The result is presented as mean ± standard deviation (*n* = 5). The hydrolysates without enzyme, Bromelain + Papain and Alcalase is presented in pooled sampling (*n* = 1).

**Figure 5 foods-12-03685-f005:**
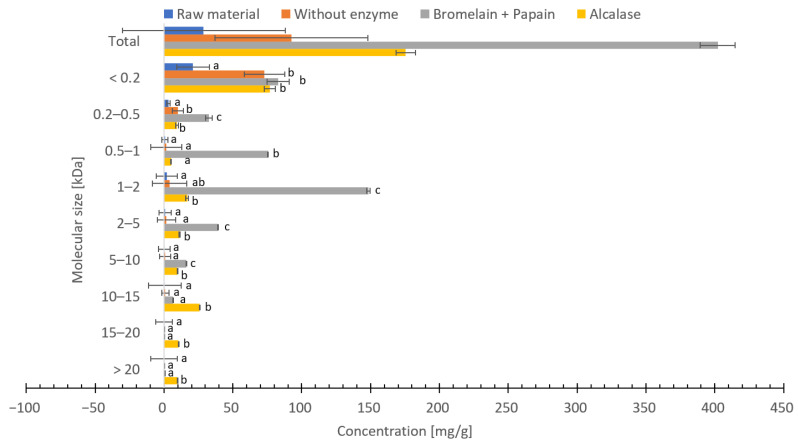
Molecular weight distribution (mg/g dry weight) of freeze-dried raw material and hydrolysates by Bromelain + Papain, hydrolysates by Alcalase or hydrolysates without enzyme of *Cucumaria frondosa*. The results are presented as mean ± standard (*n* = 5). Different letters on columns indicate significant difference between values (*p*  <  0.05).

**Figure 6 foods-12-03685-f006:**
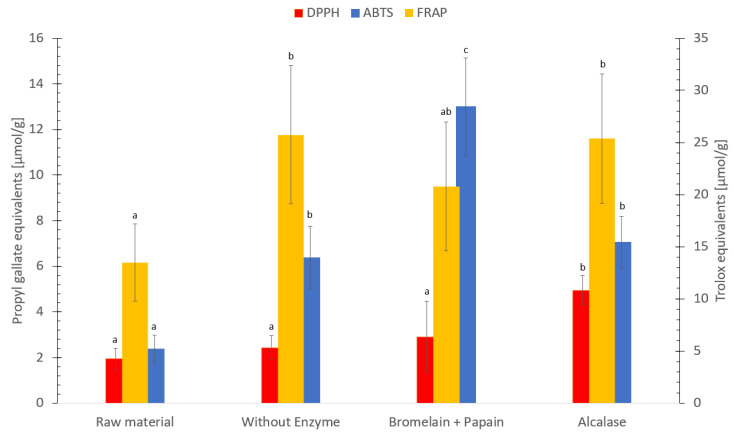
Ferric reducing antioxidant power (FRAP) measured in Trolox equivalents/g, 2′-azino-bis(3-ethylbenzothiazoline-6-sulfonic acid (ABTS) and 2,2-diphenyl-1-picrylhydrazyl (DPPH) measured in propyl gallate equivalents/g dry weight of freeze-dried raw material and hydrolysates of *Cucumaria frondosa*, protein hydrolysate without added enzyme, hydrolysate using Bromelain + Papain; hydrolysate using Alcalase. The results are presented as mean ± standard deviation (*n* = 5). Values with different letters denotes significant difference (*p*  <  0.05) samples within the same antioxidant.

**Figure 7 foods-12-03685-f007:**
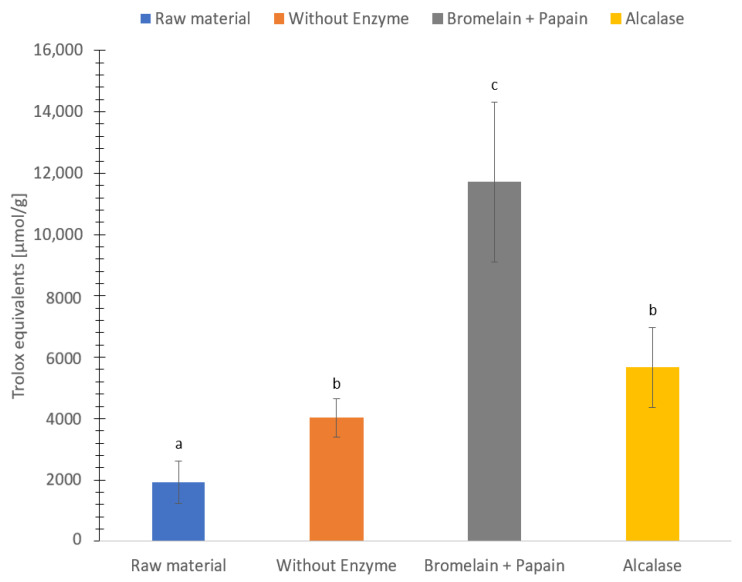
Oxygen radical absorbance capacity (µmol Trolox equivalents (TE)/g dry weight) of freeze-dried raw material and hydrolysates of *Cucumaria frondosa*, protein hydrolysate without added enzyme, hydrolysate using Bromelain + Papain, hydrolysate using Alcalase. The results are presented as mean ± standard deviation (*n* = 5). Values with different letters are significantly different (*p*  <  0.05).

**Table 1 foods-12-03685-t001:** Proximate composition of *Cucumaria frondosa* (percentage of wet weight). The results are presented as mean ± standard deviation (*n* = 5).

Sample	Water	Ash	Protein	Lipid
*C. frondosa* [%]	86.2 ± 1.5	0.6 ± 0.1	4.7 ± 0.4	0.5 ± 0.5

**Table 2 foods-12-03685-t002:** Total amino acids and free amino acids of hydrolysates of *Cucumaria frondosa*. The results are presented as mean ± standard deviation (*n* = 5). The total and free amino acids content was calculated without the water molecule.

	Total Amino Acids		Free Amino Acids	
Amino Acids	(mg/g Wet Weight)	(mg/g Dry Weight)	(mg/g Wet Weight)	(mg/g Dry Weight)
Alanine	3.5 ± 0.6	25.3 ± 4.3	0.16 ± 0.09	1.14 ± 0.66
Asparagine	0.1 ± 0.0	0.5 ± 0.1	0.00 ± 0.00	0.01 ± 0.00
Aspartic acid	5.5 ± 0.7	39.7 ± 4.9	0.01 ± 0.01	0.09 ± 0.04
Glutamic acid	6.8 ± 0.9	49.0 ± 6.7	0.33 ± 0.06	2.37 ± 0.44
Glutamine	0.1 ± 0.0	0.5 ± 0.1	0.07 ± 0.01	0.48 ± 0.08
Glycine/Arginine	5.7 ± 1.3	41.4 ± 9.4	0.32 ± 0.28	2.31 ± 2.03
Histidine	1.0 ± 0.0	7.0 ± 0.3	0.00 ± 0.00	0.01 ± 0.00
Isoleucine	2.2 ± 0.2	15.6 ± 1.1	0.00 ± 0.00	0.03 ± 0.01
Leucine	2.1 ± 0.2	15.4 ± 1.2	0.01 ± 0.00	0.07 ± 0.02
Lysine	3.8 ± 0.3	27.3 ± 2.1	0.08 ± 0.03	0.57 ± 0.25
Methionine	0.2 ± 0.0	1.6 ± 0.3	0.00 ± 0.00	0.02 ± 0.02
Phenylalanine	2.5 ± 0.2	18.2 ± 1.5	0.01 ± 0.00	0.05 ± 0.02
Serine	3.0 ± 0.4	21.5 ± 2.8	0.01 ± 0.00	0.10 ± 0.02
Threonine	2.9 ± 0.2	21.0 ± 1.7	0.01 ± 0.01	0.10 ± 0.05
Tyrosine	1.9 ± 0.2	13.4 ± 1.2	0.01 ± 0.00	0.04 ± 0.01
Valine	1.0 ± 0.1	7.4 ± 0.5	0.01 ± 0.00	0.05 ± 0.01
∑Total amino acids	42.2 ± 5.3	304.7 ± 38.3		
∑Free amino acids			1.0 ± 0.5	7.4 ± 3.7

During the process of acid hydrolysis, tryptophan is eliminated and, therefore, it is not detected. Taurine, proline and cysteine are not detected by this method. Arginine/Glycine are not separated by this method. During the process of acid hydrolysis, asparagine and glutamine underwent deamination. As a result, they are accounted for in the quantities of aspartic acid and glutamic acid, respectively.

**Table 3 foods-12-03685-t003:** Protein content (mg/g dry weight) and protein yield (% protein recovered in the aqueous phase) of hydrolysates of *Cucumaria frondosa*. The results are displayed as the mean ± standard deviation (*n* = 5). Within the same row, means that are marked with different superscript lowercase letters signify significant differences (*p* < 0.05).

Sample	Hydrolysate without Enzyme	Bromelain + Papain	Alcalase
Protein content [mg/g]	133.8 ± 35.3 ^a^	321.4 ± 52.0 ^b^	182.5 ± 43.8 ^a^
Protein yield [%]	8.3 ± 2.6 ^a^	36.8 ± 6.6 ^b^	11.8 ± 2.9 ^a^

**Table 4 foods-12-03685-t004:** Total and free amino acids (mg/g dry weight) in hydrolysates of *Cucumaria frondosa* by Bromelain + Papain (0.36 %, 1:1) or Alcalase (0.36 %). The results are presented as mean ± standard deviation (*n* = 5). Different superscript letters in the same row and within the same analysis indicate significant differences between hydrolyses (*p* < 0.05).

	Total Amino Acids			Free Amino Acids		
Amino Acids	Without Enzyme	Bromelain + Papain	Alcalase	Without Enzyme	Bromelain + Papain	Alcalase
Alanine	10.4 ± 2.1 ^a^	27.8 ± 5.8 ^b^	16.5 ± 4.5 ^a^	2.54 ± 1.73 ^a^	2.53 ± 1.11 ^a^	2.90 ± 1.57 ^a^
Asparagine	0.2 ± 0.1 ^ab^	0.3 ± 0.1 ^b^	0.1 ± 0.1 ^a^	0.01 ± 0.00 ^a^	0.07 ± 0.03 ^b^	0.02 ± 0.01 ^a^
Aspartic acid	15.8 ± 4.3 ^a^	39.4 ± 5.2 ^b^	21.9 ± 4.5 ^a^	0.17 ± 0.06 ^a^	0.13 ± 0.06 ^a^	0.36 ± 0.13 ^b^
Glutamic acid	26.1 ± 2.8 ^a^	54.3 ± 9.8 ^b^	37.7 ± 5.2 ^ab^	5.16 ± 1.71 ^a^	3.93 ± 1.36 ^a^	4.91 ± 1.24 ^a^
Glutamine	0.2 ± 0.1 ^a^	0.4 ± 0.1 ^a^	0.2 ± 0.1 ^a^	1.18 ± 0.11 ^a^	1.18 ± 0.30 ^a^	1.01 ± 0.16 ^a^
Glycine/Arginine	17.5 ± 6.8 ^a^	52.2 ± 15.2 ^b^	34.3 ± 12.9 ^ab^	4.71 ± 3.55 ^a^	4.29 ± 2.72 ^a^	4.61 ± 3.19 ^a^
Histidine	2.2 ± 0.9 ^a^	5.5 ± 0.6 ^b^	2.5 ± 0.4 ^a^	0.07 ± 0.01 ^a^	0.15 ± 0.05 ^b^	0.10 ± 0.03 ^a^
Isoleucine	5.7 ± 2.0 ^a^	12.0 ± 1.1 ^b^	6.0 ± 1.6 ^a^	0.13 ± 0.02 ^a^	0.33 ± 0.09 ^b^	0.21 ± 0.06 ^a^
Leucine	9.7 ± 3.0 ^a^	20.3 ± 2.9 ^b^	10.3 ± 3.1 ^a^	0.26 ± 0.07 ^a^	1.95 ± 0.47 ^b^	0.59 ± 0.20 ^a^
Lysine	10.0 ± 2.2 ^a^	21.8 ± 2.9 ^b^	10.1 ± 1.7 ^a^	1.18 ± 0.47 ^a^	1.28 ± 0.58 ^a^	1.30 ± 0.54 ^a^
Methionine	2.0 ± 0.7 ^a^	6.0 ± 0.5 ^b^	2.3 ± 0.6 ^a^	0.14 ± 0.03 ^a^	0.71 ± 0.16 ^b^	0.24 ± 0.09 ^a^
Phenylalanine	5.9 ± 2.2 ^a^	11.8 ± 1.4 ^b^	5.6 ± 1.6 ^a^	0.22 ± 0.05 ^a^	0.93 ± 0.21 ^b^	0.36 ± 0.11 ^a^
Serine	8.1 ± 2.0 ^a^	22.2 ± 3.5 ^b^	11.5 ± 2.4 ^a^	0.37 ± 0.06 ^a^	0.55 ± 0.15 ^a^	0.48 ± 0.13 ^a^
Threonine	8.6 ± 2.3 ^a^	19.7 ± 1.1 ^b^	10.4 ± 2.2 ^a^	0.37 ± 0.05 ^a^	0.62 ± 0.16 ^b^	0.50 ± 0.08 ^ab^
Tyrosine	4.6 ± 1.8 ^a^	10.7 ± 0.8 ^b^	4.8 ± 1.2 ^a^	0.17 ± 0.03 ^a^	0.82 ± 0.19 ^b^	0.32 ± 0.08 ^a^
Valine	6.7 ± 2.0 ^a^	17.0 ± 1.0 ^b^	8.2 ± 1.7 ^a^	0.20 ± 0.04 ^a^	0.62 ± 0.17 ^b^	0.30 ± 0.09 ^a^
∑Total amino acids	133.8 ± 35.3	321.4 ± 52.0	182.5 ± 43.8			
∑Free amino acids				16.9 ± 8.0	20.1 ± 7.8	18.2 ± 7.7

During the process of acid hydrolysis, tryptophan is eliminated and, therefore, it is not detected. Taurine, proline and cysteine are not detected by this method. Arginine/Glycine are not separated by this method. During the process of acid hydrolysis, asparagine and glutamine underwent deamination. As a result, they are accounted for in the quantities of aspartic acid and glutamic acid, respectively.

**Table 5 foods-12-03685-t005:** Protein content of raw (g/100 g wet weight and g/100 g dry weight) and hydrolyzed (g/100 g dry weight) *Cucumaria frondosa* (mean ± standard deviation, *n* = 5) as well as various conventional seafood products on the market [70].

	Unhydrolyzed Raw Material		Hydrolyzed Raw Material		
			Without Enzyme	Bromelain + Papain	Alcalase
Sample	Wet Weight	Dry Weight	Dry Weight	Dry Weight	Dry Weight
*C. frondosa*	4.2 ± 0.5	30.5 ± 0.4	13.4 ± 0.4	32.1 ± 0.5	18.3 ± 4.4
Conventional seafood products					
Protein content (g/100 g wet weight)	Tuna 19.5–26.5	Salmon 9.6–14.7	Caviar 10.0–22.3	Calamari 6.1–11.0	Shrimps 14.5–17.0

## Data Availability

The data are available upon request.

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
