# Peer review of "Enzymatic Hydrolysis of Orange-Footed Sea Cucumber (Cucumaria frondosa)—Effect of Different Enzymes on Protein Yield and Bioactivity"

_foods, 2023, doi:10.3390/foods12193685_

Round 1
Reviewer 1 Report
Comments and Suggestions for Authors
This article prepared protein hydrolysates from orange-footed sea cucumbers and determined their functions. Developing protein hydrolysates from sea cucumbers is nothing new but a continuation of previous works. However, this article provided information on free amino acids associated with the umami flavor, which could be helpful for product development. This article needs several revisions before moving to the next steps. My comments are:
Line 20, antioxidant assay should be mentioned here.
Line 50, It is commercially harvested in North America, including Canada, which should be mentioned here to provide the commercial importance.
Line 66, Why is bromelain+papain used in a combination but not Alcalase? More importantly, why were these enzymes selected while other food-grade enzymes, namely Corolase and Flavourzyme, are available? An explanation should be included.
Section 3.1, Which part of sea cucumber is used for this study? Is it the body wall? If so, how about aquapharyngeal bulbs? Were they attached to the body wall? This is very important since some body parts are considered processing waste.
Line 74, What was the particle size?
Lines 141-143, "assay" is not a part of these abbreviations.
Section 3.9, Methods to determine different antioxidant activities should be elaborated since each method has a unique mechanism of action.
Line 148, Why propyl gallate was used instead of Trolox. One should use a similar compound for a better comparison.
Section 2.9, The potential mechanism of action that shows the antioxidant activity of hydrolysates should be provided. In particular, how individual amino acids/ peptides exhibit antioxidant activity needs to be included. Authors may follow this article (https://doi.org/10.3390/md20100610), where antioxidative peptides of C. frondosa were predicted using an in silico approach.
Comments on the Quality of English LanguageMinor editing of the English language required
Author Response
Dear Reviewer,
Thank you for reviewing the article.
Please see the attachment.

Reviewer 2 Report
Comments and Suggestions for Authors
foods-2627778 review report
Enzymatic hydrolysis of orange-footed sea cucumber (Cucumaria Frondosa) – Effect of different enzymes on protein yield and bioactivity
General
The manuscript discusses the physiochemical characteristics, amino acids profiles and bioactive compound of orange-foot sea cucumber and its protein extraction by different enzymes. The authors have written a manuscript with cohesive and coherent English. However, there are still uncompleted structures and discussions that can be improved.
Introduction
Line 57 : Please briefly describe the application of bromelain, papain and Alcalase in similar studies including the optimum condition (pH, temperature, and so on) and the percentage level of the enzymes as the recommendation to be used in this study.
Result and discussion
Line 214: “Hence the Alcalase was outside its optimum temperature.” This is not a scientific excuse and shows that the author lacks information and literature when choosing the optimum conditions of protein hydrolysis of sea cucumber. Please clarify.
Line 235-236: “This could be explained by the optimum condition of the different enzymes as previously described”. See previous comments.
Author Response

(The authors gave the same response as above.)

Round 2
Reviewer 1 Report
Comments and Suggestions for Authors
The authors have significantly modified the manuscript, which can be transferred to the next steps.
Comments on the Quality of English LanguageMinor editing of English language required